# Techno-Economic Optimization of CSP Plants with Free-Falling Particle Receivers

**DOI:** 10.3390/e23010076

**Published:** 2021-01-06

**Authors:** Luis F. González-Portillo, Kevin Albrecht, Clifford K. Ho

**Affiliations:** 1Departamento de Ingeniería Energética, ETSII, Universidad Politécnica de Madrid, José Gutiérrez Abascal 2, 28006 Madrid, Spain; 2Sandia National Laboratories, Albuquerque, NM 87123, USA; kalbrec@sandia.gov (K.A.); ckho@sandia.gov (C.K.H.)

**Keywords:** particle, receiver, techno-economic, optimization, CSP

## Abstract

Particle receivers are one of the candidates for the next generation of CSP plants, whose goal is to reduce the levelized cost of electricity (LCOE) to 0.05 $/kWh. This paper presents a techno-economic analysis to study if a CSP system with free-falling particle receiver can achieve this goal. The plant analyzed integrates two ground-based bins to store the excess energy and a supercritical CO_2_ cycle to generate electricity. The model used for the analysis presents several upgrades to previous particle systems models in order to increase its fidelity, accuracy, and representativeness of an actual system. The main upgrades are the addition of off-design conditions during the annual simulations in all the components and an improved receiver model validated against CFD simulations. The size of the main components is optimized to obtain the system configuration with minimum LCOE. The results show that particle CSP systems can reduce the LCOE to 0.056 $/kWh if the configuration is composed of 1.61 × 10^6^ m^2^ of heliostats, a 250 m high tower with a 537 m^2^ falling particle curtain, and 16 h thermal energy storage.

## 1. Introduction

The goal of the next generation of concentrating solar power (CSP) plants is to produce electricity at 0.05 $/kWh [1]. In these plants, high efficiency and compact supercritical CO_2_ (sCO_2_) cycles are integrated to lower the electricity cost [2]. There are three options under investigation for the thermal carrier in the receiver: molten salt, particle, or gaseous [2]. This study analyzes if the option of particle receivers can achieve the goal of generating electricity at 0.05 $/kWh.

Particle receivers seem to be a good option since stability and non-corrosive behavior of solid particles under high temperatures [3] makes them a good fit to the sCO_2_ cycles. However, there are several challenges to address, such as the receiver design to achieve high efficiencies at an adequate cost. Thus, there are several receiver designs under investigation, mainly divided into the following groups: free-falling [4], obstructed [5], centrifugal [6] and fluidized [7]. This study integrates the free-falling particle receiver designed at Sandia National Laboratories [8].

Most of the work to date related to particle CSP systems has been focused on component development and demonstration [8,9,10]. However, techno-economic analysis of the whole system has received little attention despite it is necessary to elucidate if the levelized cost of electricity (LCOE) in these systems can achieve the goal of 0.05 $/kWh. Moreover, techno-economic analysis allows an evaluation of trade-offs, interaction between components, and identification of the optimum configuration. The results can also help to identify the size of the components needed and the general design of the system.

Techno-economic analyses of CSP systems are commonly accomplished through the NREL System Advisor Model (SAM) software [11]. SAM has several capabilities to study different types of CSP systems, such as those with molten salt [12], but particle receivers are not included. Thus, a few in-house software have been developed with the objective of analyzing the economics of particle systems [13,14,15]. Ma et al. [13] showed a first estimation of the LCOE estimation in a particle system, although the model was barely explained. Buck et al. [14] presented in 2018 a model of multi-tower CSP systems with centrifugal particle receiver to analyze the LCOE trends under different conditions. The study concluded that “the use of solid particles for solar high efficiency sCO_2_ power cycles offers unique advantages due to the wide temperature range of the particles”. However, the authors highlighted that cost models needed to be improved since there is no database for the many of the new components. Albrecht et al. [15] presented in 2019 a model in Engineering Equation Solver (EES) [16] to simulate the annual energy production of commercial scale CSP systems with free-falling particle receivers and estimate the LCOE. The results obtained with this model indicated that particle CSP systems could produce electricity below 0.06 $/kWh. The cost models of the new components were collected from previous investigations, but several improvements were needed in the model to obtain more accurate and representative results.

This paper presents several upgrades to the model from Albrecht et al. [15]. The new model can now perform annual simulations with the components working at off-design conditions during the hourly calculations. The performance at off-design conditions of power cycle is calculated with the software from Dyrevy [17] and Gavic [18], and the performance of the heliostat field is calculated by means of SolarPILOT [19]. The performance calculation of the free-falling particle receiver is upgraded by improving the calculation of optical properties and thermal losses. Moreover, the new receiver model is benchmarked against CFD simulations. Other upgrades such as the addition of parasitic loads and thermal losses in the storage are also now included in the model.

The new model is used to carry out a techno-economic optimization of CSP systems with free-falling particle receivers. The size of the main components is studied to obtain the configuration with minimum LCOE. The upgrades allow to optimize some essential parameters such as the tower height and concentration, which could not be analyzed in the previous version. Moreover, special attention is paid to the receiver the particle curtain area and the thermal energy storage (TES) size since these components must have a specific design for particle systems [20,21]. Particle loss is also analyzed due to its importance in particle CSP systems. Thanks to the new upgrades, the new model provides results with higher fidelity than previous models.

Since the new model contains several upgrades to the model from [15], its full description is presented in Section 2. The receiver model is described in special detail due to its novelty. The costs of the different components and the LCOE calculation is also described in this section. The analysis of the optimum system configuration is discussed in Section 3. The influence of the particle receiver size is analyzed in detail, together with its relation to the heliostat field and TES. Conclusions close the paper in Section 4.

## 2. Model

The particle CSP system is divided into five main parts: power cycle, primary heat exchanger, receiver, heliostat field and storage. The performance of primary heat exchanger, receiver and storage is calculated with a code developed in Engineering Equation Solver (EES). Figure 1 shows a diagram of these parts of the CSP system. The performance of the power cycle and heliostat field is calculated by means of external software. The software from Dyreby’s [17] and Gavic’s [18] theses are used for the power cycle and SolarPILOT [19] for the heliostat field.

The EES program runs annual simulations through hourly calculations. The weather data used for these calculations is USA CA Daggett (TMY2) [22]. The flowchart followed in the hourly calculations is shown in Figure 2. First, the efficiencies of heliostat field and receiver, together with the Direct Normal Irradiance (DNI) are used to obtain the power generated by the receiver if DNI and wind speed accomplish the limits shown in Table 1. The energy stored in the TES is updated, the storage thermal losses are subtracted, and the parasitic loads belonging to the receiver and heliostat field are calculated. Then, the cycle efficiency determines the thermal power required by the power cycle from the TES to supply the nominal gross electric power. Then, electricity generated, energy stored in the TES and parasitic loads are calculated. Finally, the net energy generated is calculated as the gross energy minus the parasitic loads. The models used to calculate the performance of power cycle, primary heat exchanger, receiver, heliostat field and storage are described in the next sections.

The component models and costs have been verified against actual systems and vendor quotes, where possible. However, since the model is of a conceptual next-generation CSP system, a complete validation is not possible. Moreover, the following major assumptions are considered:The output power in the power cycle is held constant at off-design conditions while pressure ratio and recompression factor are optimized to maximize thermal-to-electric efficiency.Parasitic power consumption in the power cycle is assumed to be 2% of the net power at design conditions.Cost for ground-based storage bins is assumed from Buck et al. [14], and heat losses are extracted from Sment et al. [24].Cost of piping for flow distribution in the primary heat exchanger is neglected.The solar field consists of a single tower with a north-facing receiver and a polar heliostat field located in Daggett, CA, USA.“External wind” does not affect the receiver efficiency, but the advective heat loss caused by particle flow through air is included via an advective heat transfer coefficient.Cost of horizontal conveyance of particles is neglected.Heat loss from ducts is neglected.

The presented model intends to provide a range of potential LCOE values given the inherent uncertainties of the input parameters, and it will be susceptible to changes as more data from actual systems (both in behavior and in cost) is available.

### 2.1. Power Cycle

The software developed by Dyreby [17] is used to design the recompression s-CO2 cycle. The parameters from Table 2 are used to calculate the cycle configuration with maximum cycle efficiency. The precooler needed to cool the s-CO_2_ with air at ambient temperature is designed with the software developed by Gavic [18]. The ambient temperature and the power consumed by the fan at the design point are also shown in Table 2.

Dyreby’s software is used to calculate the cycle efficiency at off-design compressor inlet temperatures by keeping the power generated constant, and Gavic’s software is used to calculate the ambient temperature required to achieve those compressor inlet temperatures with the previously designed precooler. Since cycle efficiency depends on compressor inlet temperature and the latter depends on ambient temperature, the cycle efficiency depends on the ambient temperature. Therefore, cycle efficiency will change throughout the day depending on the ambient temperature.

The ambient temperature needed to cool down the s-CO_2_ is calculated by setting the maximum fan power to 2 MW_e_ and minimum compressor inlet temperature to 32 °C. In this way, when the ambient temperature is much lower than 32 °C, the fan power required will be smaller than 2 MW_e_. Figure 3 shows the resulting cycle efficiency and fan power as a function of ambient temperature, where the cycle efficiency does not include the power consumed by the fan. The EES code uses the analytical equations fitting these results to calculate hourly cycle efficiency and fan power.

Power cycles require a startup time to ramp the component to operating temperature before producing electricity. During this time, the power cycle consumes thermal power and part of the fan power. The parameters used to define the starting point and the partial load are shown in Table 3.

### 2.2. Primary Heat Exchanger

The s-CO_2_ is heated by the particles in a moving packed-bed design heat exchanger [27]. The heat exchanger performance implemented in this system was described by [28]. The thermal power required is calculated with the cycle efficiency. S-CO_2_ inlet and outlet temperatures are determined by the power cycle configuration and particle inlet and outlet temperatures are inputs in the EES code. For the calculation of the conductance, the heat exchanger is discretized into *N_HX_* sub-heat exchangers with the same heat duty $q_{i}$, and the sum of sub-heat exchanger conductances *UA_i_* is the total conductance, *UA* [29]:(1)$$UA=\sum_{i=1}^{N_{HX}}UA_{i}=\sum_{i=1}^{N_{HX}}\frac{q_{i}}{\Delta T_{lm,i}}$$
where $\Delta T_{lm}$ is the logarithmic mean temperature difference.

The cost of the primary heat exchanger is estimated as a function of the area (see Section 2.7). Thus, it is necessary to estimate the global heat transfer coefficient *U*. Equation (2) shows the calculation of this parameter as a function of the heat transfer coefficient of particles and CO_2_ sides, $htc_{p}$ and $htc_{CO_{2}}$, respectively:(2)$$U={\left(\frac{1}{htc_{p}}+\frac{1}{htc_{CO_{2}}}\right)}^{-1}$$

Since the CO_2_ thermal resistance is much smaller than the particle thermal resistance, the dominant parameter in Equation (2) is the particle heat transfer coefficient $htc_{p}$. The calculation of the particle heat transfer coefficient is performed using a Nusselt number correlation developed for plug flow between parallel plates [30]. Converting the Nusselt correlation into a heat transfer correlation requires the particle side hydraulic diameter and effective packed-bed conductivity. The Zehner, Bauer, and Schlünder (ZBS) model described in [31] is used to estimate the packed bed conductivity, which has been shown to be in good agreement with measured values of relevant particles.

### 2.3. Thermal Energy Storage

The thermal energy storage is composed of two ground-based bins (one for the hot particles and the other for the cold ones). The energy inside the storage is measured in equivalent hours, which are defined as “the number of hours that the TES can provide energy to the power block to work at nominal power” [32]. The parameters used to define the TES are shown in Table 4.

### 2.4. Heliostat Field

SolarPILOT [19] is used to design the heliostat field. Several heliostat configurations are obtained for different values of receiver area, tower height and absorbed power. Hourly simulations along a year are carried out for all these configurations. The resulting heliostat efficiencies (i.e., the ratio between power hitting the receiver and power reaching the heliostats) are stored in tables that EES will use during the hourly calculations. Moreover, the land area calculated by SolarPILOT is also stored for the later use in EES in the cost calculation.

The parameters used in SolarPILOT to design the heliostat field are shown in Table 5 (the rest of parameters are parameters by default in SolarPILOT v1.3.8). Tower height, design power and receiver area are the parameters changing for the different configurations. The thermal losses are set to zero since these losses are simulated apart by the receiver model, i.e., SolarPILOT is only used to calculate the power hitting the receiver. Note that the tower height is defined as the distance between the heliostat pivot point and the midpoint of the receiver.

The receiver designed with SolarPILOT is a flat plate with no cavity. This could involve a slightly overestimated heliostat efficiency. To counteract this effect, the concentration is not simulated as a single aim point, but with certain distribution along the curtain (defined by the minimum image offset), which decreases the heliostat efficiency. The heat flux hitting the receiver $q_{rec}$ is defined in Equation (3), where $C_{geo}$ is the geometrical concentration (heliostat field reflective surface area divided by particle curtain surface area) and $\eta_{hel}$ the heliostat efficiency:(3)$$q_{rec}^{\prime \prime }=C_{geo}\eta_{hel}DNI$$

### 2.5. Receiver

The receiver model is the most novel part of this study. The main modifications to the receiver model from [15] are the calculation of convection and radiation losses and the calculation of the initial thickness needed to achieve a certain mass flow rate (following model proposed by [21]). Moreover, the equations used to calculate curtain absorptance and transmittance have also been upgraded.

#### 2.5.1. Energy Balance

The receiver is composed of a particle curtain with a front part and a back part, a back wall and the ambient. “The receiver is modeled using a reduced order model where the important physics are captured over a single dimension (y) in the fall direction” [15]. This model follows the same techniques as previous models of particle receivers [33,34].

Equations (4)–(6) represent the conservation equations used for the particle curtain, where $\varphi_{p}$ is the particle volume fraction, $t_{c}$ the curtain thickness, $\rho_{p}$ the particle density, $v_{p}$ the velocity, $h_{p}$ the particle specific enthalpy, $g_{c,front}$ the front part irradiance, $g_{c,back}$ the back part irradiance, $j_{c,front}$ the front part radiosity, $j_{c,back}$ the back part radiosity and $q_{adv}^{\prime \prime }$ the advection thermal loss. Equation (4) represent the mass balance, Equation (5) the momentum balance and Equation (6) the energy balance:(4)$$\frac{-d\varphi_{p}t_{c}\rho_{p}v_{p}}{dy}=0$$
(5)$$\frac{-d\varphi_{p}t_{c}\rho_{p}v_{p}^{2}}{dy}+\varphi_{p}t_{c}\rho_{p}g=0$$
(6)$$\frac{-d\varphi_{p}t_{c}\rho_{p}v_{p}h_{p}}{dy}+g_{c,front}-j_{c,front}+g_{c,back}-j_{c,back}-q_{adv}^{\prime \prime }=0$$

The irradiances $g_{c,front}$ and $g_{c,back}$ and the radiosities $j_{c,front}$ and $j_{c,back}$ are defined in Equations (7)–(10), where F is the view factor between curtain and cavity, $\varepsilon_{c}$ is the curtain emissivity, $\rho_{c}$ the curtain reflectance, $\tau_{c}$ the curtain transmittance and $j_{w}$ the radiosity coming from the back wall:(7)$$g_{c,front}=C_{geo}\eta_{hel}DNI$$
(8)$$j_{c,front}=F\left(\varepsilon_{c}\sigma T_{p}^{4}+\rho_{c}g_{c,front}+\tau_{c}g_{c,back}\right)$$
(9)$$g_{c,back}=j_{w}$$
(10)$$j_{c,back}=\varepsilon_{c}\sigma T_{p}^{4}+\rho_{c}g_{c,back}+\tau_{c}g_{c,front}$$

Optical properties reflectance and transmittance are calculated by means of the analytical model proposed by González-Portillo et al. [35].

The advection thermal loss $q_{adv}^{\prime \prime }$ is defined by means of a heat transfer coefficient called advection heat transfer coefficient whose value is discussed later. Equation (11) shows the definition of the advection thermal loss as a function of the advection heat transfer coefficient $htc_{adv}$, the particle temperature $T_{p}$ and the ambient temperature $T_{amb}$:(11)$$q_{adv}^{\prime \prime }=htc_{adv}\left(T_{p}-T_{amb}\right)$$

The energy conservation equation to the back wall is presented in Equation (12), where $k_{w}$ is the wall conductivity, $T_{w}$ the wall temperature, $g_{w}$ the back-wall irradiance, $j_{w}$ the back-wall radiosity and $q_{cv}^{\prime \prime }$ the convection loss with the ambient:(12)$$\frac{d}{dy}\left(k_{w}\frac{dT_{w}}{dy}\right)+g_{w}-j_{w}-q_{cv}^{\prime \prime }=0$$

The back-wall irradiance $g_{w}$ and radiosity $j_{w}$ are defined in Equations (13) and (14):(13)$$g_{w}=j_{c,back}$$
(14)$$j_{w}=\varepsilon_{c}\sigma T_{w}^{4}+\left(1-\varepsilon_{c}\right)g_{w}$$

The convection loss $q_{cv}^{\prime \prime }$ is defined in Equation (15), where $htc_{conv}$ is the heat transfer coefficient between back wall and ambient:(15)$$q_{cv}^{\prime \prime }=htc_{cv}\left(T_{w}-T_{amb}\right)$$

Equations (4)–(14) represent the conservation equations for each differential height (y). The code contains these equations dividing the curtain in 40 sections with the same length. The initial thickness $t_{c,0}$ is calculated with Equation (16), which is recommended by [21], where $\dot{m}$ is the mass flow rate, $w_{c}$ the curtain width, $\varphi_{p,0}$ the initial particle volume fraction (i.e., the volume fraction of a compact bed) and $d_{p}$ the particle diameter:(16)$$t_{c,0}={\left(\frac{60\dot{m}}{62w_{c}\varphi_{p,0}\rho_{p}\sqrt{g}}\right)}^{1/1.5}+1.4d_{p}$$

The thickness evolution along the fall is given by Equation (17) [36]:(17)$$t_{c}\left(y\right)=t_{c,0}+0.0087y$$

Equation (17) has a small effect on the volume fraction in comparison to the effect of gravitational acceleration.

The power gained by the particle curtain can be calculated as the enthalpy gain from the top to the bottom of the curtain. This power gained divided by the sun radiation hitting the receiver $g_{c,front}$ will be the receiver efficiency, *η_rec_*.

The thermal losses of the particle receiver are classified in radiation losses, advection losses, and convection losses. Radiation loss equals the sum of the radiosities $j_{c,front}$ times the curtain area, advection losses equals the sum of $q_{adv}^{\prime \prime },$ and convection losses equals the sum of $q_{cv}^{\prime \prime }$.

#### 2.5.2. Thermal Losses

There are two unknown parameters in the receiver model with a high importance in the receiver efficiency: the advection heat transfer coefficient $htc_{adv}$ and the view factor between curtain and cavity F. These two parameters will determine advection losses and radiation losses, respectively, and they will depend on the receiver design. The value of these parameters is obtained with CFD simulations.

CFD simulations from Mills et al. [8] are used to calculate the thermal losses of two free-falling particle receivers with different size under different conditions of power input, inlet and outlet temperatures. The parameters advection heat transfer coefficient $htc_{adv}$ and the view factor F are adjusted in the EES model to fit the thermal losses from CFD. Table 6 shows the main characteristics of the two receivers and the obtained values for advection heat transfer coefficient $htc_{adv}$ and view factor *F*. Figure 4 shows the parity plots between EES and CFD simulations for advection and radiative losses.

The view factor between curtain and cavity will mainly depend on the cavity design. For this study, the highest view factor, (*F* = 0.9) is used as reference case to consider the worst-case scenario. The advection heat transfer coefficient depends on the curtain size, specifically on the curtain length. Higher heights involve higher particle velocities, and so higher heat transfer coefficients. Thus, a correlation is developed to account for this relation considering that the advection heat transfer coefficient behaves as the convection heat transfer coefficient for a wall, where the Nusselt number is proportional to ${Re}^{6/7}$ [37]. Equation (18) shows the Nusselt number (dimensionless advection heat transfer coefficient) as a function of the Reynolds number (dimensionless velocity) obtained to fit the two advection heat transfer coefficients from Table 6, where the velocity used to calculate the Reynolds number is the velocity of particles at the bottom of the curtain and the thermophysical properties are obtained for the average temperature between the curtain average temperature and the air temperature:(18)$$Nu_{adv}=-758.9+0.05737\cdot {Re}^{6/7}$$

#### 2.5.3. Parameters

The parameters used to define the receiver are shown in Table 7. Thermophysical properties correspond to CARBO HSP sintered bauxite particles. These properties are considered constant in the temperature range of this study (~600–800 °C). The actual temperature-dependent property values (e.g., for specific heat) vary by only several percent in this temperature range according to measured data at Sandia National Laboratories. Particle emissivity is assumed to be the same as the absorptivity [38].

### 2.6. Parasitic Loads

The following parasitic loads are calculated hourly in the EES model:Cooling power in the power cycle (described in Section 2.1)Lifts in the receiver, primary heat exchanger and cold storageHeliostat drive powerFixed load

All the particle lifts are assumed to be skip hoists with efficiencies exceeding $\eta_{lift}$ = 80% [42]. The power consumed by these lifts, ${\dot{W}}_{lift}$, is calculated in Equation (19), where ${\dot{m}}_{p}$ is the mass flow rate transported, $H_{lift}$ the lift (or tower) height and *g* the gravity:(19)$${\dot{W}}_{lift}=\frac{{\dot{m}}_{p}H_{lift}g}{\eta_{lift}}$$

The parameters used to calculate the parasitic loads are shown in Table 8.

### 2.7. Costs

Previous energy models are combined with component cost models to complete the techno-economic analysis. Table 9 shows the cost models related to heliostat field, receiver, primary heat exchanger and thermal energy storage. These costs have been previously presented in [15] except tower cost from [11]. The cost models for the power cycle (except the primary heat exchanger) are shown in Table 10 and come from [43].

The cost models from Table 9 and Table 10 are used to calculate the total capital cost as shown in Equation (20), where $C_{BOP}$ is the Balance of Plant cost:(20)$$C_{cap}=C_{field}+C_{rec}+C_{HX}+C_{TES}+C_{PC}+C_{BOP}$$

The total cost of the system is the sum of direct costs and indirect costs, defined in Equations (21)–(23) where contingency $f_{contingency}$, construction $f_{construction}$ and land costs $C_{land}$ are introduced:(21)$$C_{total}=C_{direct}+C_{indirect}$$
(22)$$C_{direct}=\left(1+f_{contingency}\right)C_{cap}$$
(23)$$C_{indirect}=f_{construction}C_{direct}+C_{land}$$

From the total cost $C_{total}$, the LCOE can be calculated according to Equation (24) where *CRF* is the capital recovery factor, $OM_{fix}$ the fixed operation and maintenance cost and $OM_{var}$ the variable operation and maintenance cost:(24)$$LCOE=\frac{C_{total}\mathrm{CRF}+OM_{fix}{\dot{W}}_{\mathrm{net}}}{W_{\mathrm{elec},\mathrm{annual}}}+OM_{var}$$

The capital recovery factor *CRF* is defined in Equation (25) considering an inflation *i* during *N* years in the real discount rate *f′* from Equation (26):(25)$$CRF=\frac{f^{\prime }{\left(1+f^{\prime }\right)}^{N}}{{\left(1+f^{\prime }\right)}^{N}-1}$$
(26)$$f^{\prime }=\frac{\left(1+f\right)}{\left(1+i\right)}-1$$

The parameters used in the economic analysis are presented in Table 11. If a conventional CSP plant with molten salt receiver is simulated and optimized in SAM [11] using the presented economic model, the resulting LCOE is 0.07 $/kWh, which above the cost target of 0.05 $/kWh [1].

## 3. Results

This section presents some of the main aspects to consider in the optimization of a CSP system with free-falling particle receiver. The influence of the receiver area is analyzed due to the novelty of the model, and also due to its relevance in the LCOE. Moreover, different trade-offs study the optimum relation between heliostat field and receiver surfaces for different conditions. Tower height, solar multiple and hours of storage are also included in the analysis to achieve the minimum LCOE. The resulting LCOE depends on the particles loss during the system operation, which is analyzed at the end of the section.

### 3.1. Receiver Size

The receiver size is one of the main open questions in the configuration of particle systems. In this study, the receiver size is analyzed by means of the particle curtain area, whose value determines the receiver cost and influences the receiver efficiency. This efficiency depends on the power input as well as the particle curtain area. Radiation, advection, and convection losses determine the receiver thermal losses which, subtracted from the power input, result in the power absorbed by the curtain.

Figure 5 shows the dimensionless thermal losses (thermal loss divided by input power) of a particle receiver as a function of the curtain area. Advection losses, radiation losses and total thermal losses (i.e., 1-η_rec_) are represented in the figure for two scenarios, one with 750 MW_t_ input power and the other with half the input power, 375 MW_t_. Convection losses from the receiver back wall are not represented since they are below 1% of the input power. The scenario with the highest power input could represent the power hitting the receiver at nominal conditions for a specific heliostat field, and the scenario with the lowest power input could represent the power hitting the receiver at off design conditions (half optical efficiency or half DNI) for the same heliostat field.

The receiver efficiency is higher at greater concentrations [45], which can be appreciated in the smaller dimensionless thermal losses at a higher input power and smaller curtain areas. This fact is accentuated in particle receivers due to the increase of transmittance at bigger curtain areas. When the curtain area is small, the particles will be close enough to each other to induce an opaque curtain. However, as the curtain area increases, the curtain becomes more transparent due to the greater distance between the particles and the transmittance increases. The solar radiation gets inside the cavity, but it can easily exit too.

The increase of transmittance at bigger curtain areas is accentuated by the fact that, since the absorbed power decreases, the mass flow rate needed to keep the temperature outlet constant is lower. Then, the initial curtain thickness is smaller, and so is the entire curtain thickness. The bigger distance between particles (and so the lower volume fraction) and the smaller curtain thickness involve an exponential increase of radiation loss at big curtain areas. This can be appreciated in the case of 375 MW_t_ input of Figure 5 when the curtain area is around 1200 m^2^. The result is that receiver areas above 1300 m^2^ cannot achieve the required outlet temperature when the power input is below 375 MW_t_. This also happens when the input power is 750 MW_t_, but in much bigger curtains.

Note that Figure 5 keeps constant the input power into the receiver. However, if the same solar field were used for providing the thermal power to the different receiver areas, the input power would decrease at lower receiver areas. Thus, while smaller receiver areas involve lower thermal losses in the receiver (i.e., higher receiver efficiency), they also involve lower performance in the concentration of solar radiation (i.e., a lower heliostat efficiency).

Heliostat and receiver efficiencies are analyzed for several CSP systems with different surfaces of heliostats and receiver areas. The heliostat field configuration is optimized for each pair of heliostat field area and curtain area in order to maximize the energy reaching the receiver. The tower height for these calculations is set to 250 m. Figure 6 shows optical and receiver efficiencies at the design point defined by a DNI = 950 W/m^2^ as a function of receiver area for three different surfaces of heliostats. Concentration ratios (geometrical and optical) associated to every case are presented in Figure 7.

The heliostat field efficiency increases with the curtain size since it is easier for the heliostats to hit a bigger receiver. On the other hand, the receiver efficiency decreases due to the greater advection and radiation losses. The greatest decrease in receiver efficiency occurs in the system with heliostat area A_hel_ = 10^6^ m^2^. In this case, the input power reaching the receiver is the smallest despite the higher optical efficiency. Thus, the radiation losses increase more at larger receiver areas, as shown in Figure 5, which involve lower receiver efficiencies. Note that these efficiencies are calculated at the design point, but the efficiencies at off-design conditions (smaller power inputs) will be lower and the effect of the receiver area on the receiver efficiency will be even more noticeable.

Annual simulations are run for the CSP systems analyzed in Figure 6 and Figure 7. The hours of storage, whose influence is analyzed in detail in the next section, are set to 16 h for these calculations. The total cost of the different systems and the resulting net energy produced and LCOE are shown in Figure 8.

The shape of the net energy curve is determined by the efficiencies of heliostat field and receiver. When the curtain area is small, the net energy produced is low due to the low optical efficiency of the heliostat field. The net energy produced increases at bigger curtain areas due to the increase of optical efficiency. But since the receiver efficiency decreases, the net energy reaches a maximum when the curtain area is around 500 m^2^ (this value slightly depends on the heliostat field surface). As the curtain area increases above 500 m^2^, the net energy decreases. The slope of this decrease is greater for the case with heliostat field area A_hel_ = 10^6^ m^2^ due to the greater influence of radiation losses, as it was previously explained.

The case with the biggest heliostat area is the one with the highest net energy produced due to the greater power input into the CSP system. However, while the increase in net energy produced from A_hel_ = 10^6^ m^2^ to A_hel_ = 1.5 × 10^6^ m^2^ is around 200 GWh at the maximum point, this increase is much smaller from A_hel_ = 1.5 × 10^6^ m^2^ to A_hel_ = 2 × 10^6^ m^2^. The reason of this small increase is that, when the heliostat field area is A_hel_ = 2 × 10^6^ m^2^, the storage gets full several times along the year, and so most of the extra energy produced by this heliostat field in comparison to the heliostat field with area A_hel_ = 1.5 × 10^6^ m^2^ is wasted and cannot be converted into electricity.

The CSP system with the biggest heliostat field is also the most expensive one, and the system with the smallest heliostat field is the cheapest one. While there is a big variation in cost for the different solar field sizes, the cost hardly changes for the different curtain sizes. The cost slightly increases with area of the receiver due to the increase of the receiver size, except at small receiver areas, where the land area needed for the same heliostat surface is greater, involving a higher total cost.

The LCOE integrates the results from the net energy generated (denominator in Equation (24)) and the total cost (part of the numerator in Equation (24)). Since the total cost hardly depends on the curtain area, the LCOE shape in Figure 8 is given by the inverse of the net energy generated. The minimum LCOE is found for the second bigger solar field. The value of this LCOE is 0.056 $/kWh, and it is obtained with a curtain area A_c_ = 600 m^2^, i.e., a geometrical concentration ratio C = 2500.

Although the minimum LCOE is found for a curtain area A_c_ = 600 m^2^, areas between 500 m^2^ and 800 m^2^ can achieve LCOEs below 0.057 $/kWh. This means that the final selection of the curtain size in this range is open to be selected according to engineering constraints or tradeoffs. On the one hand, the problem of big curtains remains in the complexity of managing more particles and, on the other hand, the problem of small areas is the high concentration of radiation, which may involve high temperate gradients and high temperatures. In other types of receiver such as receivers with molten salts, the solar concentration is limited due to this latter problem. However, since particles can manage high temperatures and there are no tubes, particle receivers may allow higher concentrations.

### 3.2. System Analysis

The previous section analyzed the impact of particle curtain area and heliostat reflective surface area on the system performance and cost. In system analyses, the receiver and heliostat field are often studied indirectly integrated under two dimensionless variables: solar multiple and concentration ratio. The relation between receiver and heliostat area is analyzed by means of the concentration ratio, and the size of the solar field by means of the solar multiple. In this way, the LCOE shown in Figure 8, represented as a function of receiver area for different heliostat surfaces, can also be represented as a function of the geometrical concentration ratio for different solar multiples (see Figure 9). The minimum LCOE in Figure 9 is achieved with solar multiple SM = 3 and concentration ratio C = 2500, and its value is the same than in Figure 8, LCOE = 0.056 $/kWh.

The LCOE shape is previously justified through the relation between receiver efficiency, heliostat efficiency and cost. The variables analyzed up to now are receiver and heliostat areas. However, there is another important factor affecting heliostat and receiver efficiency and cost: the tower height. This value is set to 250 m in previous results (Figure 6, Figure 7, Figure 8 and Figure 9). The influence of tower height on the LCOE is shown in Figure 10 and Figure 11 for different solar multiples and concentration ratios.

The shape of the LCOE curves is due to two opposite effects when the tower grows: the increase of optical efficiency and the increment of the cost. At short towers, the increase in optical efficiency when the tower grows overweighs the cost increment, and at high towers, the opposite happens. The result is that there is an optimum tower height to minimize the LCOE for each pair of solar multiple and concentration ratio.

The impact of the concentration ratio is small in the range between C = 2000 and C = 3000. However, the solar multiple has a great impact in the LCOE curves. Bigger solar multiples involve bigger heliostat fields with greater difficulty of concentrating the solar radiation into the receiver. In these cases, a high tower can add extra optical efficiency compensating the greater cost. Thus, the bigger the solar multiple, the higher the optimum tower is. This optimum height is 250 m in all the cases, except for the case with solar multiple SM = 3.5. In this case, the optimum tower height is 290 m, but the LCOE is only 0.001 $/kWh lower than the LCOE obtained with a tower height H_tower_ = 250 m.

It can be observed that the shape of the LCOE is almost linear between the points at tower height 200 m, 250 m, 300 m and 350 m. The reason is that the look-up table containing the optical efficiencies calculated with SolarPILOT only includes data for these heights, and the optical efficiencies of solar fields with other tower heights are linearly interpolated. Although this may involve small errors in the calculation of the optimum tower height, the results show that this value will be 250 m, or close to it, so this height is selected for the next analysis.

The last variable to be analyzed in this system analysis is the storage size, measured in storage hours. This variable is analyzed last since the capacity to store energy in the system (set to 16 h in previous results) does not affect system efficiencies and hardly affects the shape of the curves previously studied (although it affects the absolute LCOE values).

Figure 12 shows the LCOE as a function of hours of thermal energy storage for three geometrical concentration ratios (C = 2000, C = 2500, C = 3000) and three solar multiples (SM = 2.5, SM = 3, SM = 3.5). The optimum concentration ratio is C = 2500 regardless of the solar multiple and, of course, regardless of the storage hours (since it is independent). The optimum solar multiple is SM = 2.5 if the hours of storage are below 12 and SM = 3 for the rest of cases. The optimum configuration has 16 h of storage with a solar multiple SM = 3, which results in a LCOE = 0.056 $/kWh. The main parameters of this system are summarized in Table 12.

### 3.3. Particle Loss

Previous sections consider that the particle loss in the system is neglectable. However, current investigations in the National Solar Thermal Test Facility of Sandia National Laboratories (SNL) show that on-test particle receivers have continuous particle loss. Since the particle loss can be one on the main challenges to overcome in particle receivers, its influence in the LCOE must be analyzed.

The cost of the particle loss in the system depends on the particle cost ($c_{particle}$), on the total mass circulated through the receiver along the operating life of the CSP system ($N_{life}m_{rec,annual}$) and on fraction of particle mass loss over the circulating mass (*f*_loss_), as shown in Table 9. Fraction of particle mass loss, particle cost and operating life have been previously set to 0.0001%, 1 $/kg and 30 years, respectively (see Table 11); and the total mass circulated through the receiver depends on the solar system configuration. Figure 13 shows the impact of system configuration, particle cost and fraction of particle flow loss on the LCOE. The LCOE is shown as a function of fraction of particle flow loss for three different system configurations (SM = 2.5&TES = 10 h, SM = 3&TES = 10 h and SM = 3&TES = 16 h) and two different particle costs (c = 1 $/kg and c = 0.1 $/kg). The minimum fraction of particle flow loss shown in the figure is the one used for previous calculations, 0.0001%.

When the particle cost is 1 $/kg, the three system configurations (with different solar multiple and hours of storage) show a similar tendency: the LCOE starts to increase exponentially after 0.01% fraction of particle flow loss. When the particle cost is ten times lower, 0.1 $/kg, the three system configurations show the same tendency, but in this case the fraction of particle flow loss from which the LCOE grows exponentially is around ten times bigger, 0.1%. While the maximum allowable fraction of particle flow loss highly depends on the particle cost, hardly depends on the system configuration, which is still solar multiple SM = 3 and 16 h of storage.

## 4. Conclusions

A techno-economic optimization of CSP plants with free-falling particle receiver was conducted through a model built in EES. The receiver model from [15] was upgraded to increase fidelity of results with the help of CFD simulations. Heliostat field and power cycle were introduced in the model by means of specialized software for this purpose [17,19]. After several parametric analysis, the minimum LCOE obtained is 0.056 $/kWh, which is below the first cost target of 0.06 $/kWh established by the SunShot initiative presented by the US Department of Energy [2], but above the most recent cost target of 0.05 $/kWh [1].

The system configuration required to achieve the minimum LCOE has a solar multiple of 3, a concentration factor of 2500, 16 h of storage and a tower height of 250 m. The resulting capacity factor of this plant is 0.8. While small changes in the solar multiple and tower height may involve big changes in the LCOE, small changes in the concentration factor (between 2000 and 3000) barely influence the result. In the same way, bigger storages up to 20 h barely increases the LCOE.

The minimum LCOE is found for a curtain area A_c_ = 600 m^2^. However, areas between 500 m^2^ and 800 m^2^ can achieve LCOEs below 0.057 $/kWh. The influence of the transmittance may be a factor to consider in the construction of the receiver. This study shows that big curtain areas have higher transmittance and so greater radiation losses. The model presented in this paper can be used to further analyze this matter.

The optimum system configuration is obtained for low fractions of particle flow loss. The results show that the optimum configuration barely depend on this fraction, but the value of the LCOE does depend. The fractions of particle flow loss must be below 0.01% to achieve the goal of 0.06 $/kWh. Although if the particle cost were reduced, this limit could be increased. In this way, if the particle cost were ten times lower, the goal of 0.06 $/kWh could be achieved with fractions of particle flow loss below 0.1%.

The presented model is unique to particle receiver systems and intended to analyze the potential LCOE values given the inherent uncertainties of the conceptual next-generation CSP systems. The component models and costs have been verified against actual systems and vendor quotes, where possible. Future upgrades to the system model may include upgraded cost curves and component models validated against real systems.

## Figures and Tables

**Figure 1 entropy-23-00076-f001:**
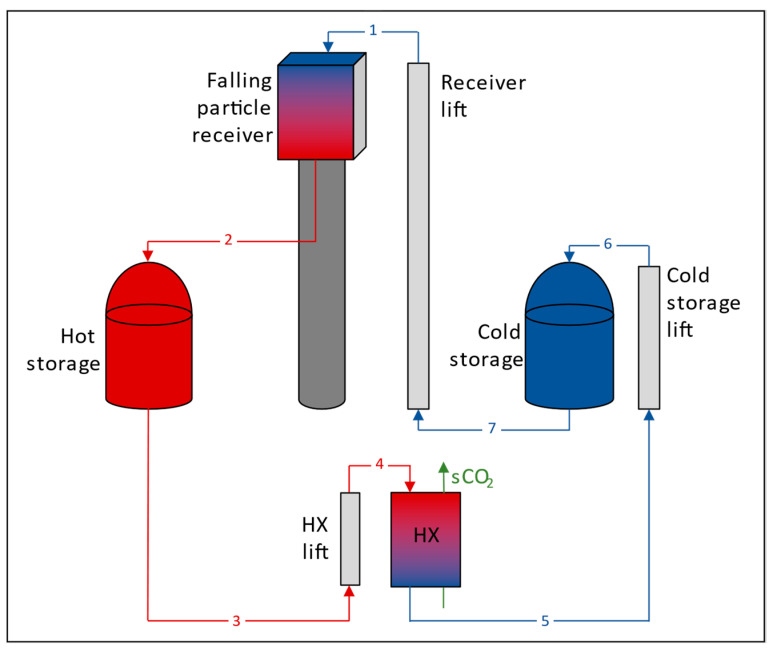
Diagram of particle CSP system coded in EES.

**Figure 2 entropy-23-00076-f002:**
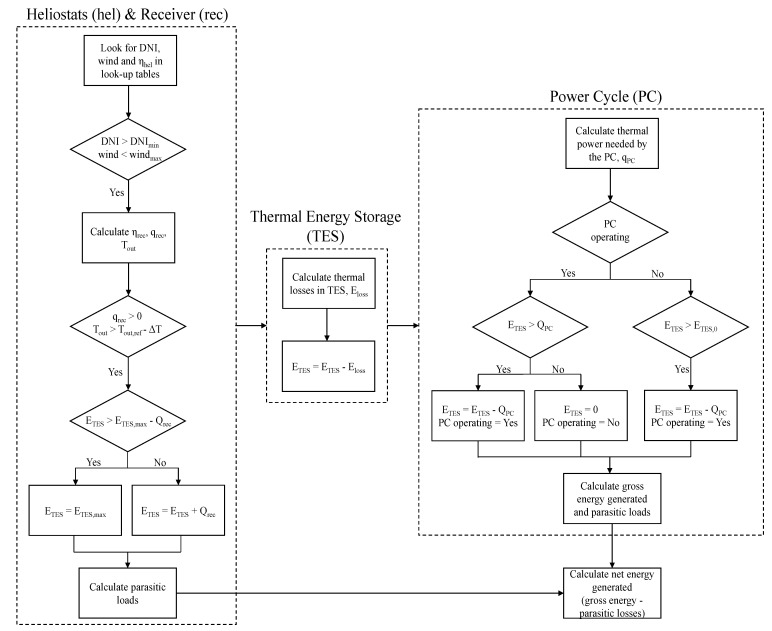
Flowchart of the hourly calculations. q: thermal power, Q: thermal energy in one hour, E: energy stored.

**Figure 3 entropy-23-00076-f003:**
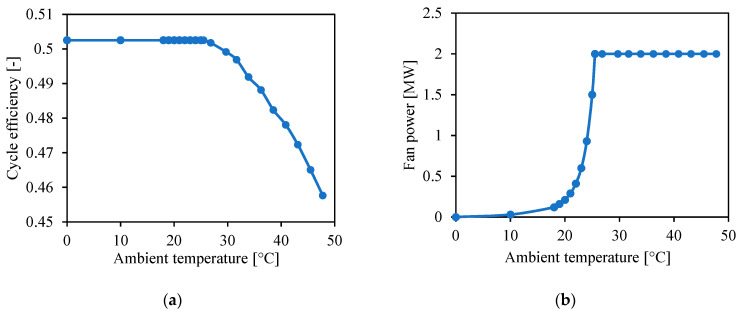
(**a**) Cycle efficiency and (**b**) fan power as a function of ambient temperature.

**Figure 4 entropy-23-00076-f004:**
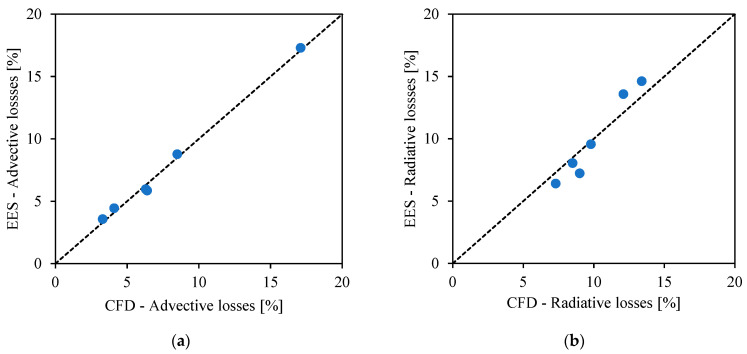
Parity plots between EES and CFD simulations for (**a**) advection losses and (**b**) radiative losses.

**Figure 5 entropy-23-00076-f005:**
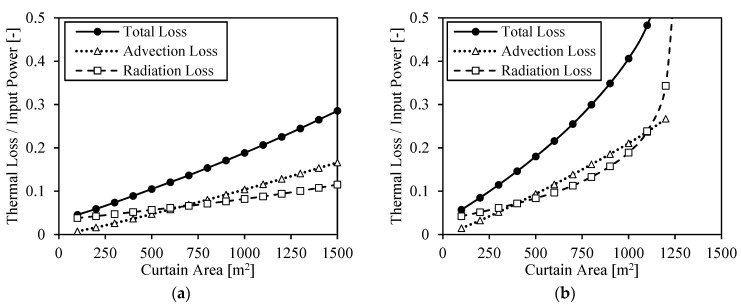
Thermal loss divided by input power as a function of the curtain area when the input power is (**a**) 750 MW_t_ and (**b**) 375 MW_t_.

**Figure 6 entropy-23-00076-f006:**
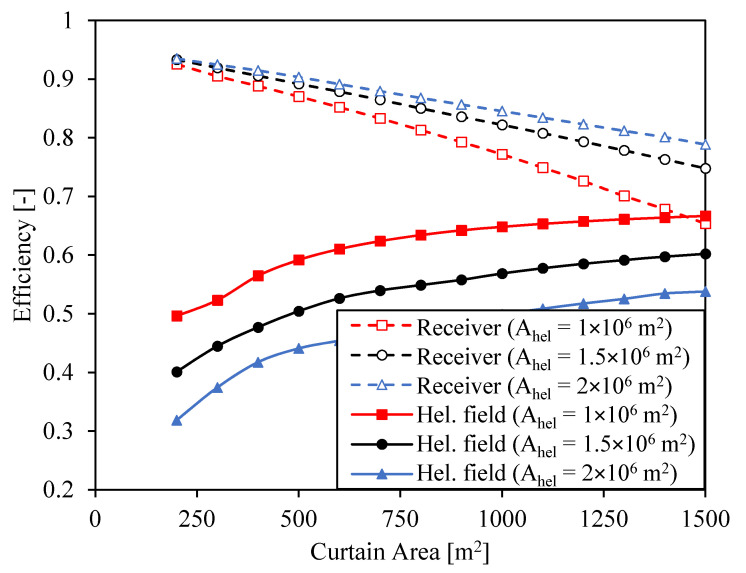
Optical and receiver efficiencies (at the design point) as a function of curtain area for three different surfaces of heliostats.

**Figure 7 entropy-23-00076-f007:**
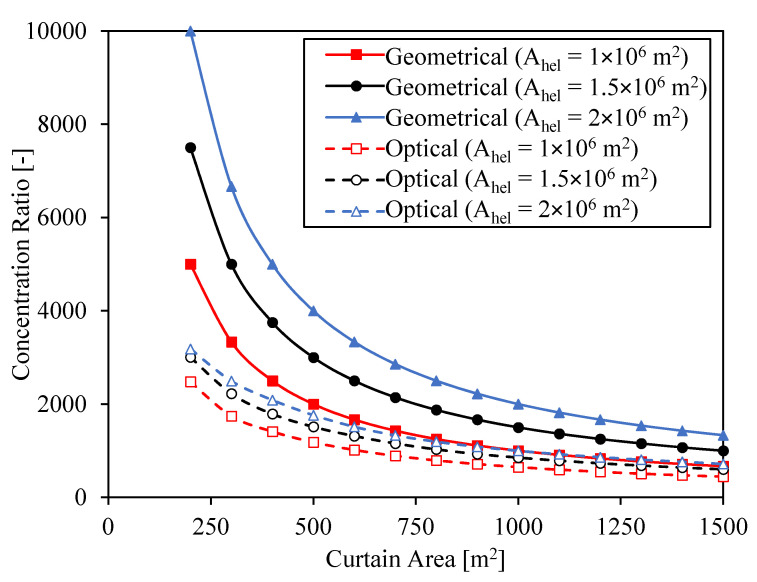
Geometrical concentration and optical concentration as a function of curtain area for three different surfaces of heliostats.

**Figure 8 entropy-23-00076-f008:**
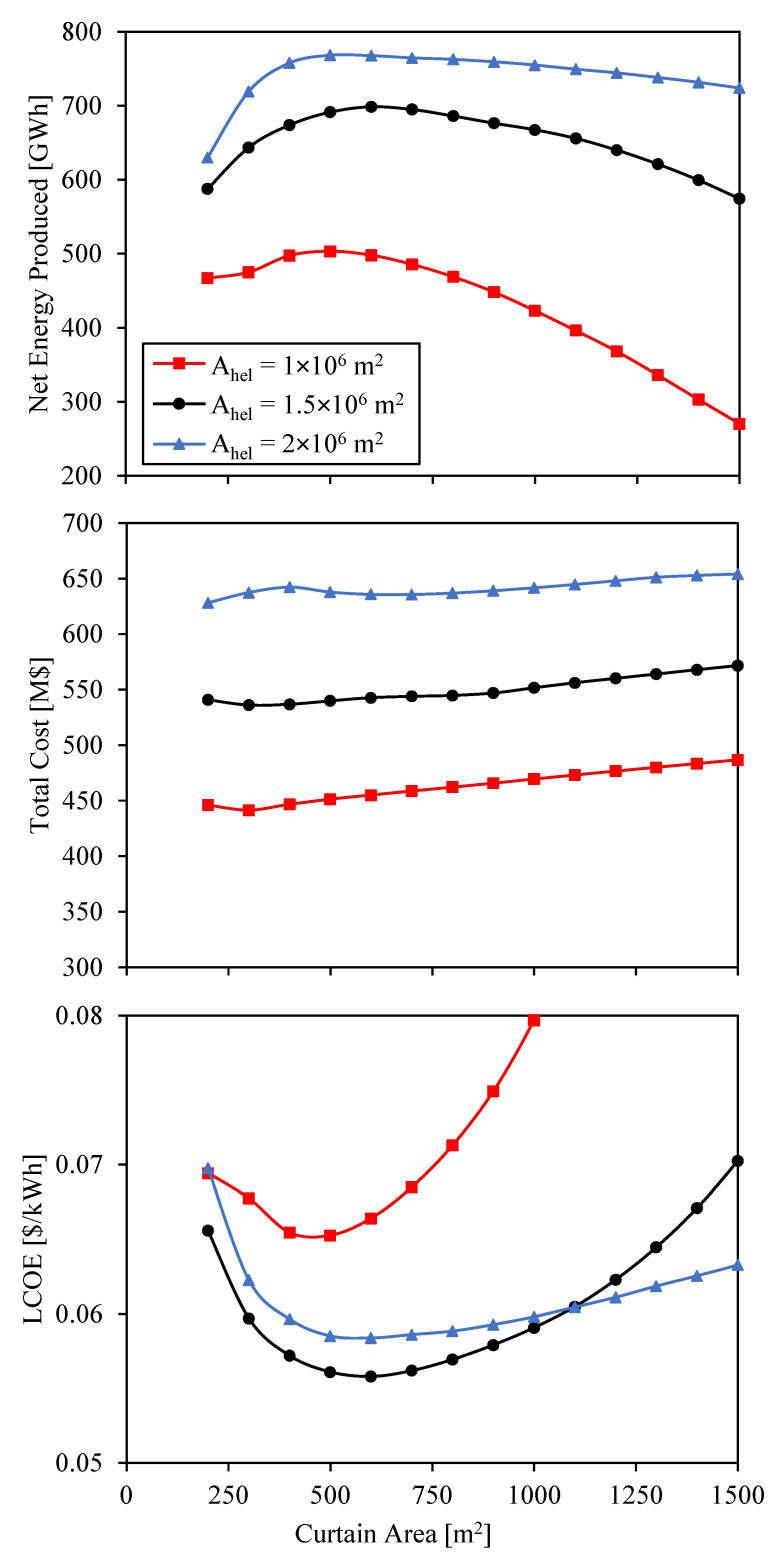
Total cost, net energy produced and LCOE as a function of curtain area for three different surfaces of heliostats.

**Figure 9 entropy-23-00076-f009:**
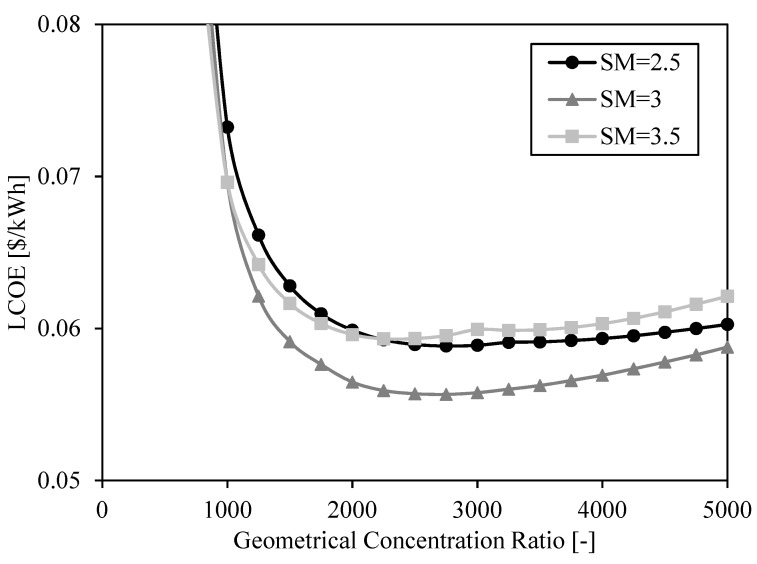
LCOE as a function of geometrical concentration ratio for three different solar multiples (SM = 2.5, SM = 3, SM = 3.5).

**Figure 10 entropy-23-00076-f010:**
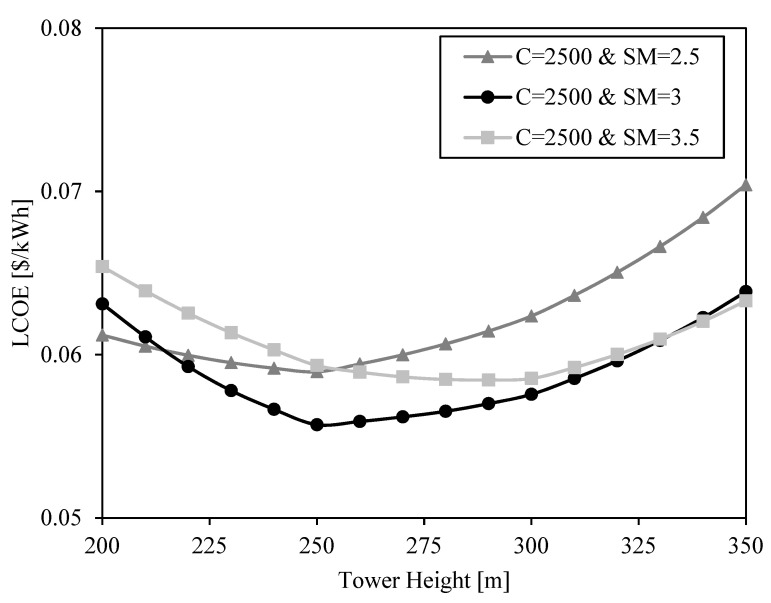
LCOE as a function of tower height for three different solar multiples (SM = 2.5, SM = 3, SM = 3.5) and a geometrical concentration ratio C = 2500.

**Figure 11 entropy-23-00076-f011:**
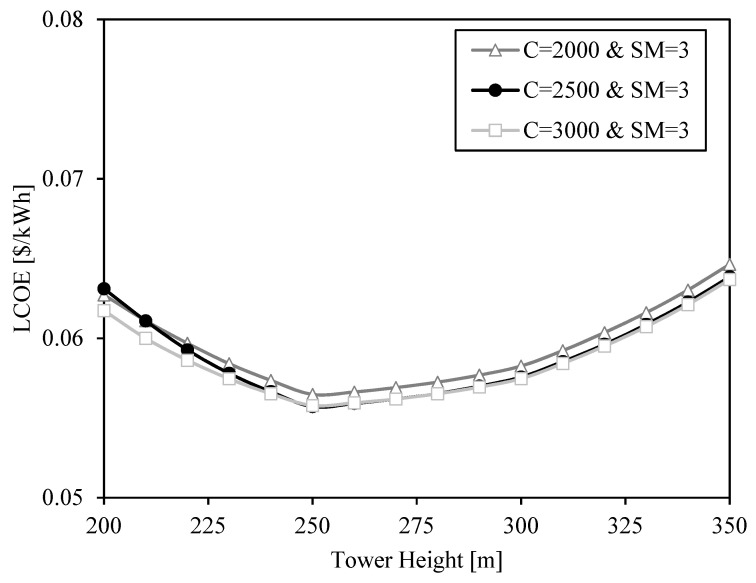
LCOE as a function of tower height for three different geometrical concentration ratios (C = 2000, C = 2500, C = 3000) and a solar multiple SM = 3.

**Figure 12 entropy-23-00076-f012:**
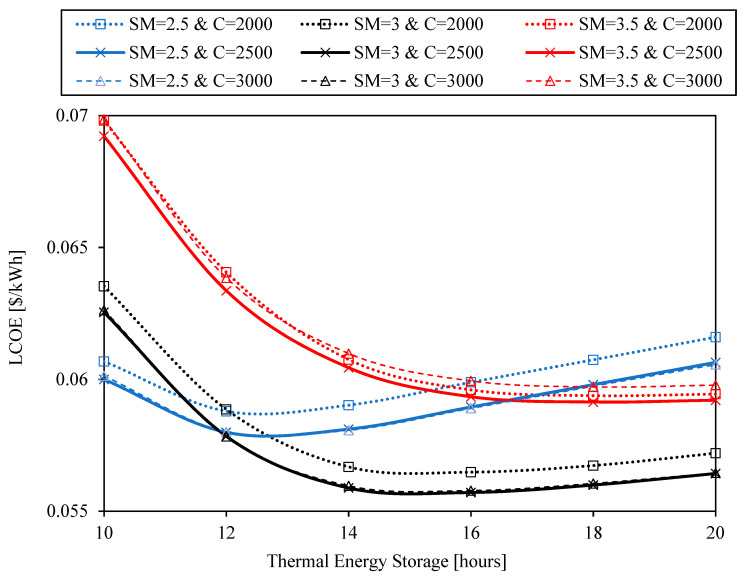
LCOE as a function of hours of thermal energy storage for three geometrical concentration ratios (C = 2000, C = 2500, C = 3000) and three solar multiples (SM = 2.5, SM = 3, SM = 3.5).

**Figure 13 entropy-23-00076-f013:**
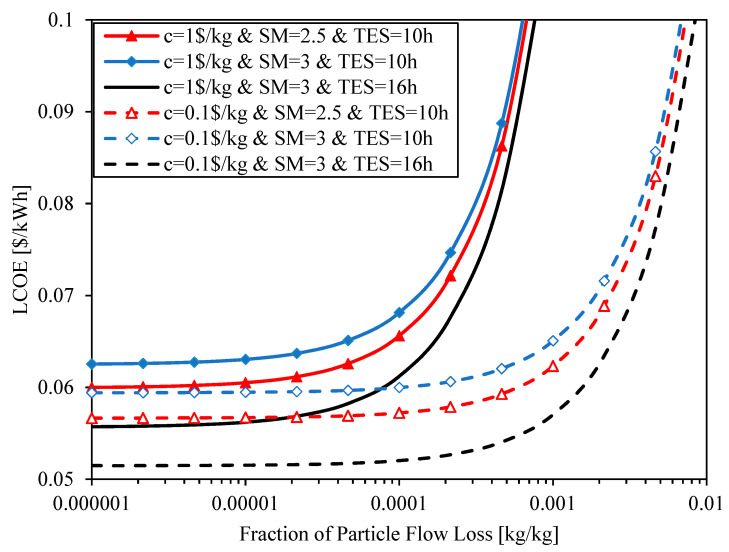
LCOE as a function of fraction of particle flow loss for configurations with different solar multiple (SM) and thermal energy storage (TES) and different particle cost (c).

**Table 1 entropy-23-00076-t001:** Weather limits.

Parameter	Value	Reference
Minimum operational DNI	200 W/m^2^	assumed to avoid inefficient operations
Maximum allowable wind speed	15.65 m/s	to avoid heliostats damage [23]

**Table 2 entropy-23-00076-t002:** Parameters used to design the power cycle.

Parameter	Value	Reference
Maximum high-side pressure	25 MPa	[25,26]
Low-side pressure	Optimized	
Recompression factor	Optimized	
Ambient temperature	35 °C	[25,26]
Compressor inlet temperature	41 °C	[26]
Turbine inlet temperature	700 °C	[26]
Compressor isentropic efficiency	0.8	[26]
Turbine isentropic efficiency	0.87	[26]
Relative pressure drop in every heat exchanger	0.005	[25,26]
Minimum temperature difference in LTR	5 °C	assumed
Minimum temperature difference in HTR	5 °C	assumed
Fan power	2 MW_e_	[25,26]
Net power	100 MW_e_	[23]
Estimated gross to net conversion factor	0.9	[11]

**Table 3 entropy-23-00076-t003:** Parameters used for partial load.

Parameter	Value
Power cycle start time	0.5 h
Minimum storage energy to start power cycle, *E_TES,0_*	3 h
Fraction of fan power during start time	0.5

**Table 4 entropy-23-00076-t004:** Parameters to define the thermal energy storage.

Parameter	Value	Reference
Aspect ratio (height/diameter)	2	assumed in [15]
Thermal resistance (conduction and convection)	4 m^2^·K/W	[24] ^1^

^1^ The thermal resistance was extracted from the insulation design and bin geometry given by [24] where the cyclic charge and discharge process was modeled to determine heat loss.

**Table 5 entropy-23-00076-t005:** Parameters used to design the heliostat field.

Parameter	Value
Tower height	Variable (200–350 m)
Design power	Variable (200–1200 MW)
Receiver height	Variable (10–50 m)
Receiver width	Receiver height
Solar field extent angle	+/− 90°
Maximum field radius	25
Receiver type	Flat plate
Thermal losses	0 W
Receiver thermal absorptance	0
Minimum image offset	3

**Table 6 entropy-23-00076-t006:** Characteristics of the simulated receivers.

Parameter	Small Scale	Commercial Scale
Curtain width	1.52 m	15.6 m
Curtain fall height	2.25 m	17.2 m
Aperture area	1.5 m^2^	158.9 m^2^
Distance aperture-curtain	0.4 m	3 m
Advection heat transfer coefficient	38 W/m^2^·K	95 W/m^2^·K
View factor between curtain and cavity	0.65	0.9

**Table 7 entropy-23-00076-t007:** Parameters used to define the receiver.

Parameter	Value	Reference
Receiver outlet temperature	800 °C	[15,39]
Aspect ratio	1	[40]
Back wall emissivity	0.8	assumed
Back wall thickness	0.05 m	assumed
Back wall conductivity	0.2 W/m·K	assumed
Back wall heat transfer coefficient	10 W/m^2^·K	assumed
Particle diameter	350 μm	[40]
Particle specific heat	1.2 kJ/kg·K	[41]
Particle conductivity	2 W/m·K	[41]
Particle density	3550 kg/m^3^	[41]
Maximum solid volume fraction	0.6	[21]
Particle absorptivity	0.87	[35]
Particle emissivity	0.87	estimated with [38]

**Table 8 entropy-23-00076-t008:** Parameters used to define the parasitic loads.

Parameter	Value	Reference
Lift efficiency	0.8	[42]
Heliostat tracking power	0.055 kW	[11]
Fraction of rated gross power consumed all times	0.0055 kW/kW	[11]

**Table 9 entropy-23-00076-t009:** Cost models related to heliostat field, receiver, primary heat exchanger and thermal energy storage. Extracted from [15] except tower cost from [11].

Component	Cost Model
Heliostat field	$C_{\mathrm{field}}=\left(c_{\mathrm{hel}}+c_{\mathrm{prep}}\right)A_{\mathrm{field}}$
Receiver	$C_{\mathrm{rec}}=C_{\mathrm{fpr}}+C_{\mathrm{tower}}+C_{\mathrm{lift},\mathrm{rec}}$
Falling particle receiver	$C_{\mathrm{fpr}}=37400\left[\frac{\$}{m^{2}}\right]A_{\mathrm{ap}}$
Tower	$C_{\mathrm{tower}}=3000000\left[\$\right]\times e^{0.0113\left[m^{-1}\right]\left(H_{\mathrm{tower}}-\frac{H_{rec}}{2}+\frac{\sqrt{A_{hel}}}{2}\right)}$
Lift	$C_{\mathrm{lift}}=58.37[\frac{\$\;\cdot \;s}{m\cdot \mathrm{kg}}]h_{\mathrm{lift}}{\dot{m}}_{p}$
Primary heat exchanger	$C_{\mathrm{HX}}=\sum A_{\mathrm{HX},i}c_{\mathrm{HX},i}$
Primary heat exchanger (cost per m^2^)	$c_{\mathrm{HX}}=\{\begin{array}{c} 1000\left[\frac{\$}{m^{2}}\right]\\ 1000\left[\frac{\$}{m^{2}}\right]+0.3\left[\frac{\$}{m^{2}\cdot {{}^{\circ}C}^{2}}\right]{\left(T_{p,\mathrm{in}}-600\left[{}^{\circ}C\right]\right)}^{2} \end{array}$ $\begin{array}{l} T_{p,\mathrm{in}}<600{}^{\circ}C\\ T_{p,\mathrm{in}}\geq 600{}^{\circ}C \end{array}$
Storage	$C_{TES}=c_{bin,hot}A_{bin,surf}+c_{bin,cold}A_{bin,surf}+C_{lift,HX}+C_{lift,cold}+C_{p}+C_{p,loss}$
Bin	$c_{\mathrm{bin}}=1230\left[\frac{\$}{m^{2}}\right]\left(1+0.3\frac{T-600}{400}\right)$
Particles	$C_{p}=\left(1+NS\right)c_{p}m_{p,TES}$
Particle loss	$C_{p,loss}=N_{\mathrm{life}}c_{p}m_{\mathrm{rec},\mathrm{annual}}f_{\mathrm{loss}}$

**Table 10 entropy-23-00076-t010:** Cost models related to power cycle [43].

Component	Cost Model
Power cycle	$C_{PC}=C_{c}+C_{\mathrm{rc}}+C_{t}+C_{HTR}+C_{LTR}+C_{cooler}+C_{gen}$
Compressor and recompressor	$C_{c}=1230000\left[\frac{\$}{{\mathrm{MW}}^{0.3992}}\right]{\dot{W}}_{c}{}^{0.3992}$
Turbine	$C_{t}=182600\left[\frac{\$}{{\mathrm{MW}}^{0.5561}}\right]{\dot{W}}_{t}{}^{0.5561}f\left(T_{max}\right)$ $f\left(T_{max}\right)=\{\begin{array}{c} 1\\ 1+1.106\times 10^{-4}{\left(T_{max}-550\left[{}^{\circ}C\right]\right)}^{2} \end{array}$ $\begin{array}{l} T_{max}<550{}^{\circ}C\\ T_{max}\geq 550{}^{\circ}C \end{array}$
Regenerator (HTR and LTR)	$C_{reg}=49.45\left[\frac{\$\cdot K^{0.7544}}{W^{0.7544}}\right]UA^{0.7544}f\left(T_{max}\right)$ $f\left(T_{max}\right)=\{\begin{array}{c} 1\\ 1+1.137\times 10^{-5}{\left(T_{max}-550\left[{}^{\circ}C\right]\right)}^{2} \end{array}$ $\begin{array}{l} T_{max}<550{}^{\circ}C\\ T_{max}\geq 550{}^{\circ}C \end{array}$
Cooler	$C_{cool}=32.88\left[\frac{\$\cdot K^{0.75}}{W^{0.75}}\right]UA^{0.75}$
Generator	$C_{gen}=108900\left[\frac{\$}{{\mathrm{MW}}^{0.5463}}\right]{\left({\dot{W}}_{t}-{\dot{W}}_{c}-{\dot{W}}_{rc}\right)}^{0.5463}$

**Table 11 entropy-23-00076-t011:** Parameters used in the economic analysis.

Parameter	Value	**Reference**
Heliostat Cost, $c_{\mathrm{hel}}$	75 $/m^2^	[23,44]
Site Preparation, $c_{\mathrm{prep}}$	10 $/m^2^	[15,44]
Land cost, $c_{land}$	2.47 $/m^2^	[23,44]
Particle Cost, $c_{p}$	1 $/kg	Estimated cost of bulk CARBO HSP 40/70
Non-Storage Inventory, *NS*	5%	assumed
Particle Loss, $f_{\mathrm{loss}}$	0.0001%	assumed
Balance of Plant Cost, $c_{BOP}$	167 $/kW_e_	[11,23] ^1^
Contingency Cost, $f_{contingency}$	10%	[23]
Construction Cost, $f_{construction}$	9%	[23]
Discount Rate, *f*	7%	[23]
Inflation, *i*	2.5%	[23]
Fixed O&M Cost, $OM_{fix}$	40 $/kW·year	[23,44]
Variable O&M Cost, $OM_{var}$	0.003 $/kWh	[23]
Lifetime, *N*	30 years	[23]

^1^ The BOP cost is calculated as the ratio ‘BOP cost/power block cost’ from SAM [11] times the power block cost estimated by DOE in [23].

**Table 12 entropy-23-00076-t012:** Parameters of the optimum CSP system configuration.

Parameter	Value
Solar multiple	3
Concentration ratio	2500
Tower height	250 m
Storage hours	16 h
Heliostat surface	1.61 × 10^6^ m^2^
Curtain area	537 m^2^
Storage bin volume	m^3^
Capacity factor	0.82
Total cost	560 M$
Annual net energy generated	720 GWh
LCOE	0.056 $/kWh

## Data Availability

The data presented in this study are available on request from the corresponding author.

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
