# Peer review of "Techno-Economic Optimization of CSP Plants with Free-Falling Particle Receivers"

_entropy, 2021, doi:10.3390/e23010076_

Round 1

Reviewer 1 Report

All the acronyms should be presented the first time are introduced, e.g. Engineering Equation Solver (EES).

If possible, the Authors could provide some information about the variability of the particles parameters as a function of the temperature.

Some missing units are present. E.g. line 335.

I suggest to add markers to the lines in the figures with results in order to show for what values the simulation were carried out, as made for Fig. 9 and so on.

Attached a version of the paper with some marked issues.

Author Response

Thank you very much for your response and comments. Please find enclosed the fully revised version of our manuscript. This version contains some minor changes according to your suggestions.

  1. All the acronyms should be presented the first time are introduced, e.g. Engineering Equation Solver (EES).

Done.

  1. If possible, the Authors could provide some information about the variability of the particles parameters as a function of the temperature.

Table 7 shows particle properties. A sentence is added to clarify that the properties are considered constant in the temperature range of this study (~600 °C – 800 °C):

“The parameters used to define the receiver are shown in Table 7. Thermophysical properties correspond to CARBO HSP sintered bauxite particles. These properties are considered constant in the temperature range of this study (~600 °C – 800 °C). The actual temperature-dependent property values (e.g., for specific heat) vary by only several percent in this temperature range according to measured data at Sandia National Laboratories”

  1. Some missing units are present. E.g. line 335.

The sentence has been clarified:

“Convection losses from the receiver back wall are not represented since they are below 1% of the input power”

  1. I suggest to add markers to the lines in the figures with results in order to show for what values the simulation were carried out, as made for Fig. 9 and so on.

Thanks for the comment. Figures 5-8 show now for what values the simulations were carried out.

  1. Attached a version of the paper with some marked issues.

Thanks. The issues have been solved.

Reviewer 2 Report

Second round of Review:

Authors answered all the raised issues and questions from the reviewers, especially those related to scope, originality, model validation and state-of-the-art. I therefore think that it is suitable for publication.

Detailed remarks herinbelow.

1) Scope of the article

Authors answered the former first comment:

"The scope of this article is within the scope of the Special Issue "Thermodynamic Optimization of Complex Energy Systems" (https://www.mdpi.com/journal/entropy/special_issues /Thermodynamic_Optimization). Anyway, if the Editor believes this topic is not valid for Entropy, the authors can send it to another journal."

Ok, my mistake. I was not aware of this special issue.
This optimization is more system-related and economic-related, rather than pure thermodynamics (that is why I concuded that it was not in the scope of the journal) but it completely fits in the specifications that are given by the journal for this special issue (sCo2 & CSP)

2) Self citation and previous study by the author

Authors included their first study about this topic. However, it has been filtered by the journal, and I still do not have access to it. From its abstract, the slide presentation that I found and the different answers to the reviewers, I can assume that this new paper is sufficiently different from the former one.

On the self-citation topic, authors gave an acceptable answer, and added some descriptions of other approaches that are similar to theirs => OK

3) Minor corrections

Page 1, l. 25-26
"In these plants, high efficiency and compact supercritical CO2 (sCO2) cycles are integrated to lower the electricity cost [2]."

I would rather prefer "can be used", because other cycles can be used (Brayton cycle with air...)

Page 1 l. 38
"if the levelized cost of electricity (LCOE) in these systems can achieve the goal of generating electricity at 0.05 $/kWh"

I would shrink it to "the goal of 0.05 $/kWh"

Page 2 line 64
"which increase its feasibility"

I would remove this last part which, to my opinion, is confusing

Page 2 line 75 to 80
"Since the new model contains several upgrades to the model from [15], its full description is presented in Section 0. The receiver model is described in special detail due to its novelty. The costs 77 of the different components and the LCOE calculation is also described in this section. The analysis of the optimum system configuration is discussed in Section 0. The influence of the particle receiver size is analyzed in detail, together with its relation to the heliostat field and TES. Conclusions close the paper in Section 0."

Please fix the sections' links

Table 1
"Maximum allowable wind speed 15.65 m/s [23]"

Would it be possible to quickly give the reason of this maximum, before the reference?

Page 6, line 151
"Table 3."

Please remove the "carriage return" before "table"

Table 3
"Storage needed for stating power cycle, ETES,0"

starting. And I would rather write "Minimum energy to start power cycle" or something equivalent ("storage needed" appears weird)

p6 line 162
"(see section 0). Thus"

links

p8 line 213
"particle enthalpy,"

"particle specific enthalpy" is more appropriate

P9 line 232
"diving"

dividing?

Table 6
"View factor"
Please remind which view factor

p1, L 261
"The view factor will"

Same comment.

Table 7 (and others)
Please be consistent between using the dot or not within the units. I would recommend to use standard notations (W.m-2.K-1, etc.)

Table 9 (and other tables and equations)
Please be consistent between using dots for multiplications, or not

Table 11
Please use SI unit for landcost ($/km²)

p13 l. 335
"since they are below 0.01."

missing unit or specification of the share (1% of output power for example)

Figure 5
Please remove the correction marks

Table 12
Please provide all the parameters that were subject to optimization, even though some are given within the text : SM, receiver height, design power.

Cost and energy:

Why not using G$ and GWh for better readability?

"Net energy generated"

please specify "(annual)"

Author Response

Thank you very much for your response and comments. Please find enclosed the fully revised version of our manuscript. This version contains some minor changes according to your suggestions.

Page 1, l. 25-26
"In these plants, high efficiency and compact supercritical CO2 (sCO2) cycles are integrated to lower the electricity cost [2]." I would rather prefer "can be used", because other cycles can be used (Brayton cycle with air...)

Next generation of CSP plants integrates sCO2 cycles according to the cited reference:

Mehos, M.; Turchi, C.; Vidal, J.; Wagner, M.; Ma, Z.; Ho, C.; Kolb, W.; Andraka, C.; Kruizenga, A. Concentrating Solar Power Gen3 Demonstration Roadmap. NREL/TP-5500-67464 2017

Other types of cycle could be used, but the future in the mentioned document ad many others focus on sCO2 cycles for the next future.

Page 1 l. 38
"if the levelized cost of electricity (LCOE) in these systems can achieve the goal of generating electricity at 0.05 $/kWh" I would shrink it to "the goal of 0.05 $/kWh"

Thanks for the suggestion. It has been accepted.

Page 2 line 64
"which increase its feasibility" I would remove this last part which, to my opinion, is confusing

Thanks for the suggestion. It has been accepted.

Page 2 line 75 to 80
"Since the new model contains several upgrades to the model from [15], its full description is presented in Section 0. The receiver model is described in special detail due to its novelty. The costs 77 of the different components and the LCOE calculation is also described in this section. The analysis of the optimum system configuration is discussed in Section 0. The influence of the particle receiver size is analyzed in detail, together with its relation to the heliostat field and TES. Conclusions close the paper in Section 0." Please fix the sections' links

Thanks. It has been fixed.

Table 1
"Maximum allowable wind speed 15.65 m/s [23]" Would it be possible to quickly give the reason of this maximum, before the reference?

The reason is to protect the heliostats against damage from strong winds. It has been added to the table.

Page 6, line 151
"Table 3." Please remove the "carriage return" before "table"

Thanks. It has been fixed.

Table 3
"Storage needed for stating power cycle, ETES,0" starting. And I would rather write "Minimum energy to start power cycle" or something equivalent ("storage needed" appears weird)

Thanks for the suggestion. It has been accepted.

p6 line 162
"(see section 0). Thus" links

Thanks. It has been fixed.

p8 line 213
"particle enthalpy," "particle specific enthalpy" is more appropriate

Thanks for the suggestion. It has been accepted.

P9 line 232
"diving" dividing?

Thanks. It has been fixed.

Table 6
"View factor" Please remind which view factor

Thanks, it has been clarified in several parts of the paper.

p1, L 261
"The view factor will" Same comment.

Thanks, it has been clarified in several parts of the paper.

Table 7 (and others)
Please be consistent between using the dot or not within the units. I would recommend to use standard notations (W.m-2.K-1, etc.)

Thanks for the comment. The issue has been addressed.

Table 9 (and other tables and equations)
Please be consistent between using dots for multiplications, or not

Thanks for the comment. The dot has been maintained just in one specific case to improve the understating.

Table 11
Please use SI unit for landcost ($/km²)

The units have been changed as suggested.

p13 l. 335
"since they are below 0.01." missing unit or specification of the share (1% of output power for example)

The sentence has been clarified:

“Convection losses from the receiver back wall are not represented since they are below 1% of the input power”

Figure 5
Please remove the correction marks

Thanks for the comment. The issue has been solved.

Table 12
Please provide all the parameters that were subject to optimization, even though some are given within the text : SM, receiver height, design power.

Done

Cost and energy:

Why not using G$ and GWh for better readability?

Thanks for the comment. M$ and GWh are now used.

"Net energy generated" please specify "(annual)"

Done

This manuscript is a resubmission of an earlier submission. The following is a list of the peer review reports and author responses from that submission.

Round 1

Reviewer 1 Report

The topic of the proposed manuscript is interesting, well written and it is suitable with the scope of Entropy Journal. However, there are several points of concern that should be addressed before eventual publication.

Below some specific points of concern explaining my position. Attached

The Authors should pay attention to the use of an adequate terminology (e.g. bins).

Putting references in the abstract should be avoided.

Typically, the description of the system configuration in the abstract is put before the results.

The keywords should be improved (e.g. particle…receivers) and increased for a better visibility of the paper.

The results section should begin with a brief summary of the analyzes carried out.

In the first part of introduction, all the three options for next-gen CSP should be introduced.

The reference for 0.06 $/kWh misses. Why such value is assumed as reference? It is a universal value for all the word? Economically competitive prices of producing electrical energy with RES depend on many factors.

The review regarding the research topic is too short. It is suggested to expand this part.

Specify the specialized software mentioned in the introduction.

It is not correct to state that the model is validated with a CFD model, because it is not a validation. Comparison is a better term in this case.

The novelty of the paper should be highlighted.

The first external software is validated? Some more details could be added about this tool.

All the acronyms should be defined the first time are introduced in the text.

The range of the used parameters should be stated instead of only the information that they are variable.

Why the thermal losses are set to zero in the heliostat field design?

A scheme of the receiver should be added to understand better the proposed model.

The detailed information how the CFD simulation and the comparison were carried out should be added.

The parameter section is not clear in some parts (second sentence).

The assumptions adopted in the model should be discussed as well as the its limits.

In my opinion, it is not necessarily to put units in the cost function equations.

Reviewer 2 Report

This article deals with a techno-economical analysis of a new type of CSP plant that is promising in terms of LCOE reduction. It is well-written and provides sufficient information to the readers in order to be able to reproduce its results.

However, I have several concerns about its relevancy to be published in the Entropy journal:

1) Scope

I am not sure that the article falls within the scope of the journal, which is:

"

Entropy deals with the development and/or application of entropy or information-theoretic concepts in a wide variety of applications. Relevant submissions ought to focus on one of the following:

- develop the theory behind entropy or information theory
- provide new insights into entropy or information-theoretic concepts
- demonstrate a novel use of entropy or information-theoretic concepts in an application
- obtain new results using concepts of entropy or information theory "

"

There is not exergy or entropy analysis in this work.

2) Self-citations and other works on related topics

Author 3 has 15/39 self-citations
+1 self-citations of author 1 alone
=> 41% of self-citations. Even if not a lot of people are working on falling particle receivers, isn't it too much?

Falling particle receivers are one of the new technologies that are investigated around the world to improve solar towers. Other teams are working on centrifugal receivers or fluidized bed particles, that are close (in my opinion) in terms of ideas. I think that the introduction should at least explain the different areas that are currently explored.

3) Method

It misses a validation of the model (the physical one, but most of all the economical one). Maybe it is in the first reference of the paper, however, I cannot find any access to it.

Therefore:

- I cannot check how close or distinct are the two works;

- I cannot check how the economical model has been validated.

Without validation, how can we be sure that the LCOE of a conventionnal solar tower plant would not fit below the 0.06$/kWhe? The model should be applied to other configurations in order to confirm or disprove the potential of falling-particle receivers.

Please provide me a pdf of this proceeding, if possible.

Few minor corrections:

In the authors:
"Luis F. González‐Portillo1 1,*, Kevin Albrecht 2 and Clifford Ho 2"
Two '1' for the first author

Line 104:
"32ºC"
Missing Space

Line 141:
"[#]"
?

Equation 26:
Use of "*" for multiplicator
Please be consistent

Table 6:
"0.2 W/m‐K"
Please be consistent in unit notations. "-" can be misinterpretated as a minus

Table 10:
"Inflation, i 2.5 % [18]"
Be consistent with space or not before %

Line 306:
"i.e,, 1 ‐ ηrec"
Two comas

Figure 4:
Please reming somewhere what is "tau" and the difference between black and red (in the description for example), as it is done in the text
Please check the font of the legend
Please change the format so that the x-axis is not cut

Line 344:
"DNI = 950 m2"
Please correct the unit

Line401:
"Overheating of the cavity walls, especially near the receiver, may still be an issue for particle receiver."
This sentence has no link with the previous ones

Line 432:
"C = 2000 and C= 3000."
Please be consistent with spacing throughout the whole text

Line 452:
Same comment. Choose between putting spaces before and after all "=", or never putting them

Table 11:
"Heliostat surface 1.58ˑ106 m2"
Please be consistent between the use of "E" and "10^"